# The Small GTPase Arf6: An Overview of Its Mechanisms of Action and of Its Role in Host–Pathogen Interactions and Innate Immunity

**DOI:** 10.3390/ijms20092209

**Published:** 2019-05-05

**Authors:** Tim Van Acker, Jan Tavernier, Frank Peelman

**Affiliations:** 1Cytokine Receptor Laboratory, Flanders Institute of Biotechnology, VIB-UGent Center for Medical Biotechnology, Faculty of Medicine and Health Sciences, Ghent University, 9000 Ghent, Belgium; Tim.VanAcker@vib.be (T.V.A.); Jan.Tavernier@vib-ugent.be (J.T.); 2Department of Biomolecular Medicine, Ghent University, B-9000 Ghent, Belgium

**Keywords:** Arf6, small GTPase, immunity, host–pathogen interaction, vesicular transport, innate immunity, toll-like receptor, phagocytosis

## Abstract

The small GTase Arf6 has several important functions in intracellular vesicular trafficking and regulates the recycling of different types of cargo internalized via clathrin-dependent or -independent endocytosis. It activates the lipid modifying enzymes PIP 5-kinase and phospholipase D, promotes actin polymerization, and affects several functionally distinct processes in the cell. Arf6 is used for the phagocytosis of pathogens and can be directly or indirectly targeted by various pathogens to block phagocytosis or induce the uptake of intracellular pathogens. Arf6 is also used in the signaling of Toll-like receptors and in the activation of NADPH oxidases. In this review, we first give an overview of the different roles and mechanisms of action of Arf6 and then focus on its role in innate immunity and host–pathogen interactions.

## 1. Introduction

### 1.1. Arf6 and Its Relatives

The ADP-ribosylation factor (Arf) protein family is part of the large Ras superfamily which encompasses small GTPases [1]. Arfs received their name for their assistance in the ADP-ribosylation of the α-subunit of heterotrimeric G proteins induced by the cholera-toxin [2]. A plethora of cellular processes require Arf proteins. The human Arf family includes Arf-like proteins (Arls), Sar proteins, and six Arfs (Arf1–Arf6). The group of 6 Arf proteins is subdivided into three classes based on sequence homology: class I (Arf1–3), class II (Arf4–5) and class III (Arf6) [3]. Class I and II Arfs are regulators of vesicle trafficking between the Golgi and endoplasmatic reticulum but also have roles outside the Golgi. Arf6, the single class III member, is both in sequence and in function the outcast of the family and is active in both clathrin-mediated endocytosis (CME) and clathrin-independent endocytosis (CIE) and in endosomal traffic and endosomal recycling outside the Golgi apparatus (Figure 1) [4]. Arf6 is located at the plasma membrane, cytosol, and endosomal membranes and is N-terminally myristoylated [5]. It associates with membranes in a GTP-dependent manner; a dissociation from the plasma membrane requires GTP hydrolysis on Arf6 [6].

### 1.2. Activation and Inactivation of Arf6 through GEFs and GAPs

The activation and inactivation of Arf6 are regulated by guanine nucleotide exchange factors (GEFs) and GTPase-activating proteins (GAPs). GEFs contain a catalytic Sec7 domain which accelerates the dissociation of GDP from the Arf, allowing the insertion of GTP in the Arf nucleotide binding pocket and switching Arfs to an activated state [7,8,9]. GAPs stimulate GTP hydrolysis on Arfs through their conserved ArfGAP zinc finger domain, although ELMOD1 and 2 are an exception to this rule as they act as Arf6 GAPs without a typical ArfGAP domain [10,11,12,13]. Currently, we have knowledge of 8 Arf6 GEFs and up to 15 Arf6 GAPs. Specific Arf6-mediated processes require distinct Arf6 GAPs/GEFs for their regulation. For a detailed description and extensive overview of the Arf6 GAPs and GEFs, the reader is referred to two reviews in this area [14,15]. It should be noted that the term activation/inactivation can be misleading, as many—if not most—Arf6 mediated processes probably require the Arf6 GTP/GDP cycle rather than just the Arf6-GTP form. As Arf6 functions via a regulated cycle of GTP binding and hydrolysis, both GAPs and GEFs are necessary for its functionality [15].

### 1.3. Research Tools to Study Arf6

Functional studies of Arf6 frequently use the following tools: (i) the silencing or gene disruption of Arf6, (ii) the overexpression of Arf6 or of its GAPs or GEFs; (iii) Arf6 GEF inhibitors; (iv) the overexpression of Arf6 mutants with altered GTP/GDP binding acting as constitutively active or dominant negative forms—The “constitutively active” **Arf6-Q67L** mutant lacks GTPase activity and is associated with GTP [16]. **Arf6-T27N** is a “dominant negative” mutant that has a decreased nucleotide binding and is often presented as a mimic of the GDP-bound state [16]. However, both GTP and GDP binding are affected, and the lack of nucleotide binding can cause denaturation [17]. **Arf6-T44N** has a low affinity for GTP but retains a near-normal binding of GDP, and Arf6-T44N was, therefore, presented as a better alternative for the GDP-bound state [17]—and (v) another tool used in functional studies is a **myristoylated Arf6 peptide** corresponding to the N-terminal helix of Arf6, which can be transfected in cells by coupling it to penetratin [18,19,20]. The peptide acts as an antagonist of Arf6, possibly via competitive binding to Arf6 interaction partners, although the exact targets of this peptide have not been defined (to our knowledge).

### 1.4. Roles and Mechanisms of Arf6

#### 1.4.1. Arf6 Directly Activates Lipid Modifying Enzymes

Arf6 exerts its effects through a diverse set of effectors. It activates the lipid modifying enzymes phosphatidylinositol 4-phosphate 5-kinase (**PIP5K**) and phospholipase D (**PLD**), which lead to a production of phosphatidylinositol-4,5-bisphosphate (PIP_2_) and phosphatidic acid (PA) (Figure 2A). An increase in PA concentration will lead to a secondary enhancement of PIP_2_ production due to a stimulatory effect on PIP5K [21,22,23,24,25,26,27,28]. The Arf6-mediated increase in PIP_2_ is one of its crucial mechanisms of action in clathrin-dependent and -independent endocytosis, endosomal recycling, plasma membrane remodeling, and actin polymerization.

#### 1.4.2. Arf6 Stimulates Actin Polymerization

Arf6-GTP can also recruit GEFs or GAPs for other GTPases as effectors. This is exemplified in the role of Arf6 in actin polymerization (Figure 3): The accumulation of Arf6-GTP and PIP3 in the plasma membrane leads to a recruitment of ARF nucleotide binding site opener (**ARNO**), a GEF for both Arf6 and Arf1 [29]. ARNO will amplify Arf6 activation and additionally activate Arf1. Activated Arf1 cooperates with activated Rac1 to recruit and activate the WAVE regulatory complex (WRC), which in turn leads to **actin polymerization** through the activation of the Arp2/3 complex (Figure 3) [29,30,31]. This cooperative process is likely promoted by the binding of ARNO to the dock180/ELMO complex, which acts as Rac1 GEF [32]. Arf6 further assists actin polymerization and remodeling by the directed transport of lipid rafts, Rho GTPases, and Rho signaling components to sites of actin polymerization and by generating PIP_2_, which acts on a multitude of actin modulating proteins (Figure 3) [33,34,35,36,37].

#### 1.4.3. Arf6 and Rab35 Are Mutual Antagonists that Cooperate

Arf6-GTP binds the Rab GAP EPI64B to inhibit **Rab35**, another important small GTPase in endosomal transport and recycling (Figure 2B) [38]. Conversely, Rab35-GTP binds the Arf6 GAP ACAP2 to inhibit Arf6 (Figure 2B) [39]; hence, Arf6 and Rab35 are mutual antagonist. Moreover, Rab35 activates the lipid phosphatase OculoCerebroRenal syndrome of Lowe (**OCRL**), which hydrolyzes PIP_2_ to PIP, further antagonizing the effects of Arf6 [40,41]. While antagonists, it is now clear that Rab35 and Arf6 are both sequentially required in similar processes in the cell, such as endocytic recycling (Figure 2B) [39].

#### 1.4.4. Arf6 Can Direct Transport along Microtubules

Another important group of Arf6 effectors are the scaffold proteins **JIP3**, **JIP4**, **FIP3,** and **FIP4**, which link cargo vesicles to dynein and kinesin motor proteins for transport along microtubules (Figure 2C) [42,43]. When JIP3 or JIP4 bind Arf6-GTP, they release kinesin and bind the dynactin complex, in turn linked to dynein, reversing the direction of transport along the microtubule [42]. The Arf6 GTP/GDP cycle, thus, determines the direction of transport along microtubules. This JIP3/4-mediated transport has been shown to be important for cytokinesis, clathrin-mediated endocytosis, and the exocytosis of matrix metaloproteases [42,44,45]. FIP3 and FIP4 associate with Arf6 and Rab11, both linked to endosomal recycling, and interact with kinesin or dynein [46,47].

#### 1.4.5. Arf6 Directs Transport after Clathrin-Mediated Endocytosis and Clathrin-Independent Endocytosis

Arf6 is involved in clathrin-mediated endocytosis (CME), clathrin-independent endocytosis (CIE), and endosomal recycling (Figure 1) [4,16,44,48,49]. In both forms of endocytosis, Arf6-GTP is not strictly required for the endocytic process itself but is crucial for the downstream transport and recycling of cargo [44,48]. In CME, Arf6 is activated by dynamin-2 and bound to the adapter AP-2 in clathrin-coated pits. This Arf6-GTP contributes to PIP_2_ generation, which is required for CME [44]. Arf6 recruits the nucleoside diphosphate kinase Nm23-H1 as a GTP source for fission by dynamin in endocytosis [50]. Upon clathrin uncoating, Arf6-GTP binds JIP3 and JIP4, which mediate fast recycling [44]. 

Arf6 mediates a clathrin-independent endocytosis (CIE) pathway, where cargo is first internalized and transported in Arf6-enriched vesicles and subsequently can be recycled to the plasma membrane [4,51]. Arf6-GTP hydrolysis to Arf6-GDP and the removal of PIP_2_ are required for a fusion of these vesicles with the sorting endosomes, and the GTP-locked Arf6-Q67L mutant, therefore, accumulates in aberrant-stalled and -aggregated peripheral vesicles enriched in PIP_2_ [38,52]. Early in CME, Rab35 interacts with endocytosed vesicles, recruiting OCRL, which hydrolyzes PIP_2_ [38,41]. Rab35 also seems to act on CIE vesicles, probably leading to Arf6 inactivation and PIP_2_ hydrolysis [53]. The antagonistic Arf6 and Rab35, thus, cooperate sequentially in the internalization and early sorting process (Figure 2B).

Arf6 is involved in the fast and slow recycling of endosomal cargo from CME and CIE back to the plasma membrane. Fast recycling transports cargo directly from the sorting endosomes to the plasma membrane (Figure 1) [51]. To enable the fast recycling of cargo such as transferrin, both Arf6 GTP hydrolysis and Rab35-GTP are required [54].

During slow recycling, cargo is transported from sorting endosomes to the endocytic recycling compartment(s) (ERC), where it can be transported back to the plasma membrane via Rab8- and Rab11-dependent pathways (Figure 1) [51,55]. Arf6-T27N has a more perinuclear distribution and co-localizes with the constitutive active Rab8-Q67L, which is found on tubules emanating from the endocytic recycling compartment involved in recycling [56]. While the presence of Arf6-T27N in recycling compartments led to the hypothesis that Arf6-GDP to Arf6-GTP activation in the ERC is required for recycling, this contrasts the observation that Arf6-T44N is not present in the ERC but located at the plasma membrane [17]. Rab11 and Arf6-GTP both bind to FIP3 and FIP4, and an Arf6/FIP3/Rab11 complex is shown to mediate at least one form of slow endosomal recycling (Figure 2C) [43,46,57]. FIP3 and FIP4 are linked to dynein, which ensures transport towards the minus end of the microtubule and, thus, towards the endocytic recycling compartment, which is located close to the microtubule organizing center [46]. However, the site-specific phosphorylation of FIP3 can induce a motor switch from dynein to kinesin 1, reversing the direction of transport towards the plasma membrane (Figure 2C) [47,58]. At the plasma membrane, Arf6-GTP interacts with Sec10 of the vesicle-tethering **exocyst complex**, which may be required for a proper targeting of the recycling cargo (Figure 2C) [43,59]. 

#### 1.4.6. Arf6 Assists Autophagy

Arf6 plays a role in **autophagy**, and co-localizes with proteins that are essential for the initiation of autophagosome formation, i.e., the formation of pre-autophagic structures or phagophores [60,61]. Especially phagophore proteins Atg12 and Atg16L1 displayed a co-localization with Arf6 on autophagic vesicles [60]. Arf6 knockdown decreased the number of Atg16L1- and Atg12-positive vesicles, implying a role for Arf6 in phagophore formation in both HeLa and CHO cells. The Arf6 activation of PIP5K contributes to autophagy, as PIP_2_ influences the membrane delivery to forming phagophores by regulating plasma membrane endocytosis. Furthermore, an Arf6 mutant defective in PLD activation (Arf6-N48R) decreased the levels of LC3-II (a protein involved in the elongation and fusion of autophagosomes), suggesting a positive regulatory role for Arf6-induced PLD activation in autophagy [60].

### 1.5. Roles of Arf6 in Health and Disease

As it acts in actin polymerization, membrane remodeling, internalization, recycling, and targeted transport of receptors and other cargo, Arf6 affects health and disease in many ways [4,5,15]. It has essential roles in cell division, cell migration, cell spreading, cell polarity, and adhesion, and aberrant Arf6 activation is found in metastasis and cancer cell invasion [62,63]. Furthermore, Arf6 regulates exocytosis or secretion of diverse cargos, including GLUT1, synaptic vesicles, and gastric acid [64,65,66]. Arf6-mediated vesicle transport to the midbody is essential in cytokinesis [42,43]. Arf6 affects signaling via β-catenin, mTORC1, MAP kinases, and other pathways [67,68]. In this review, we provide an overview of the different roles of Arf6 in host–pathogen interactions and innate immune processes.

## 2. Arf6 Drives Phagocytosis through Multiple Mechanisms of Action, which Can Be Counteracted by Pathogens

Phagocytosis is crucial for the innate immune system as it leads to the resolution of infection by removing pathogens and by clearing dead cells. The phagocytic process initiates by the binding of a particle (e.g., an opsonized pathogen) to specific cell surface receptors (e.g., Fcγ receptors (FcγR) or complement receptors) on a professional phagocyte, such as macrophages or neutrophil [69]. The receptor–particle interaction leads to an intracellular activation of small GTPases of the Rho family such as Rac1, Cdc42, and RhoA, which drive actin polymerization. The formation of new actin filaments generates pseudopod protrusions of the cell surface able to surround the captured particle, referred to as the phagocytic cup (Figure 4). Upon closure of the phagocytic cup, the particle is engulfed by the cell and captured in a large vesicular structure, called a phagosome. The newly formed phagosome matures by interacting with other endosomal compartments and will eventually fuse with a lysosome enabling the degradation of the phagosome content [70].

Arf6 was first implicated in phagocytosis in 1998 when Zhang et al. demonstrated that an overexpression of Arf6-Q67L or Arf6-T27N inhibits the FcγR-mediated phagocytosis of IgG-coated erythrocytes and attenuated the F-actin accumulation beneath these particles [71]. The participation of Arf6 in Rac1-mediated membrane ruffle formation and actin polymerization (Figure 1) is probably crucial for pseudopod extension [72]. Arf6 is also responsible for the focal delivery of endosomal vesicles to the plasma membrane during FcγR-mediated phagocytosis [73]. Exocyst components accumulate at the phagosome, and the Arf6-exocyst interaction may be implicated in the targeted delivery of vesicles to the site of phagocytosis [74]. Furthermore, the Arf6 activation of PIP5K seems important, as PIP_2_ and PIP5Kα are located at sites of phagocytosis, and phagocytosis is blocked by inhibition or dominant negative mutants of PIP 5-kinase and by hydrolysis or a blockade of PIP_2_ [75,76]. GTP-cycling-deficient Arf1 mutants also block phagocytosis upon initiation of the pseudopod extension. FcγR-mediated phagocytosis, thus, requires the normal GDP/GTP cycling of both Arf1 and Arf6 [77]. Beemiller et al. studied the location and activation kinetics of Rac1, Cdc42, Arf1, and Arf6 in phagocytosis [77,78]. Arf6 is activated immediately upon particle binding, with the peak of active Arf6 levels reached within 2 minutes of phagocytosis initiation and active Arf6 being especially localised at the leading edge of the phagocytic cup, resembling the profile of Cdc42 activation. Rac1 and Arf1 show a broader distribution over the phagocytic cup and the activation of Rac1 is reached after 5.5–7.5 min. The sealing of phagosomes requires the removal of PIP2 and likely requires Arf6 inactivation. A PI3K inhibitor (LY294002) did not inhibit Arf6 activation in FcγR-mediated phagocytosis but reduced the activation of Arf1 and prevented the closure of phagosomes [77]. The dependency of Arf1 activation on PI3 kinase activity is possibly linked to the role of PIP3 in ARNO recruitment (Figure 1) [79].

A depletion of the Arf6 GEF **BRAG2** was shown to inhibit the phagocytosis of opsonized zymosan and IgG beads, phagocytosed by complement receptor 3 (CR3) and FcγR. [80]. BRAG2 depletion suppressed F-actin recruitment around phagocytosed particles, and the overexpression of a BRAG2 mutant lacking the catalytic Sec7 domain (responsible for Arf6 activation) in BRAG2-depleted cells did not restore phagocytic activity. The overexpression of Arf6-Q67L in BRAG2-depleted cells did rescue phagocytic activity, which was not the case for Arf6-T27N overexpression. Arf6 and BRAG2 also briefly co-localized during the phagocytosis of opsonized zymosan and IgG beads, strongly suggesting that BRAG2 indeed regulates Arf6 activation during the phagocytic process [80]. The CR3-mediated phagocytosis of zymosan is also affected by cytohesin-1, another Arf6 GEF. However, this GEF has opposite effects on phagocytosis compared to BRAG2: Cytohesin-1 silencing or inhibition enhanced CR3-mediated zymosan phagocytosis in PLB-985 cells or polymorphonuclear cells [81].

The Arf6 GAP **ASAP2** was shown to accumulate at sites of phagocytosis, and its overexpression reduces FcγR-mediated phagocytosis [82]. An efficient phagocytosis may require the Selenoprotein K-mediated palmitoylation of ASAP2, which leads to its proteolytic cleavage by calpain-2 [83]. Arf6 deactivation during phagocytosis is also regulated by the Arf6 GAP **ACAP2**, recruited by Rab35 [69,84]. Rab35 was shown to be recruited early to the phagocytic cup base during the phagocytosis of IgG-erythrocytes and zymosan, in turn recruiting ACAP2. The Rab35 silencing or overexpression of GTP-locked or GDP-locked Rab35 mutants reduced the rate of phagocytosis, inhibited actin polymerization at the site of phagocytosis, and blocked pseudopod extension. A constitutively GTP-bound Rab35 mutant enhanced ACAP2 recruitment to phagosomes, and the co-overexpression of ACAP2 and GTP-locked Rab35 had a synergistic inhibitory effect on phagocytosis [69]. In this context, the Rab35 recruitment of OCRL to hydrolyze Arf6-induced PIP_2_ (Figure 2B) may also be of importance, as a knockout of the orthologue of OCRL in the amoeba *Dictyostelium discoideum* inhibits phagosome closure [85].

## 3. Bacteria and Protozoa Hijack Arf6 for Invasion in Host Cells and Virulence

The gastrointestinal pathogens enteropathogenic and enterohemorrhagic *Escherichia coli* (**EPEC** and **EHEC**) tightly associate extracellularly with the enterocyte surface to colonize the intestine. EPEC and EHEC use a type III secretion system, a needle-like protrusion from the bacterial cell wall that delivers bacterial virulence proteins into host cells. One of these type III secretion system effectors, **EspG**, blocks phagocytosis of the pathogens by interfering with the Arf6/Arf1/Rac1/WRC pathway of actin polymerization normally used in early stage phagocytosis (Figure 5A) [86]. EspG binds both Arf1 and Arf6 and blocks the interaction of Arf6 with ARNO, antagonising ARNO recruitment, which is normally required for a subsequent Arf1 activation. EspG’s interaction with Arf1 additionally inhibits the Arf1/Rac1-mediated binding and activation of the WRC (Figure 5A).

The promoting role of Arf6 in phagocytosis can have undesired side effects. The intestinal epithelium provides a barrier against the passage of commensal gut bacteria. However, this barrier function becomes compromised upon treatment with interferon-γ (IFN-γ). Transcytosis through gut epithelial cells by a noninvasive *Escherichia coli* strain is enhanced by IFN-γ, which activates extracellular signal-regulated protein kinase 1/2 (ERK1/2). ERK1/2, in turn, enables the activation of Arf6, which was proposed to assist the uptake of *Escherichia coli*, possibly by promoting transcytosis [87].

Many pathogens developed strategies to deliberately invade eukaryotic cells or to cross epithelial barriers. Several of these strategies are similar to mechanisms used in phagocytosis and often target Arf6. In this section, we provide an overview of the bacteria targeting Arf6 to initiate intracellular uptake.

***Salmonella* Typhimurium** is a facultative intracellular pathogen that causes diarrheal diseases. It ensures its entry into intestinal epithelial cells by injecting virulence factors such as **SopB** and **SopE** via a type III secretion system. In the host cytosol, these virulence proteins induce actin polymerization, leading to membrane ruffling, which facilitates the bacterial uptake. SopE is a Rho GEF that activates Rac1, thereby inducing recruitment of the WAVE regulatory complex (WRC) to the cell membrane (Figure 5B). This is insufficient for *Salmonella* entry, as Arf1 activation is required for a further activation of the WRC to induce actin polymerization and membrane ruffling [29,88]. Arf6 and its GEFs BRAG2 and EFA6 are, therefore, recruited to the site of *Salmonella* invasion, leading to an accumulation of Arf6-GTP (Figure 5B). SopB is a phosphatydylinositide phosphatase that indirectly induces the generation of PIP3. PIP3 and Arf6-GTP lead to a recruitment of ARNO, which activates Arf1. Rac1-GTP and Arf1-GTP cooperate to activate the WRC and to establish actin polymerization. Thus, Arf6 can enhance the invasion of *Salmonella* by indirectly promoting the activation of Arf1 (Figure 5B) [29,88]. 

The Arf6 GAPs **ACAP1** and **ADAP1** and the Arf1 GAP ASAP1 are found at *Salmonella*-induced ruffles, and a loss of ACAP1, ADAP1, or ASAP1 impaired *Salmonella* invasion. This could be rescued by a fast GDP/GTP cycling mutant of Arf1 or Arf6 demonstrating that, like in several other Arf6-mediated processes, *Salmonella* uptake requires GDP/GTP cycles [89]. Upon entry into host cells, the bacteria reside in a membrane-bound compartment, the *Salmonella*-containing vacuole (SCV) where they establish a replicative niche [90]. During infection, Arf6 was shown to be present on the *Salmonella*-induced host cell membrane ruffles and co-localized with the SCV 10 minutes postinfection. This co-localization disappeared 60 minutes postinfection, and Arf6 may, therefore, have further functions at the SCV, which have not yet been established [90].

***Shigella*** species are Gram-negative bacteria that invade intestinal epithelial cells and subsequently cross intercellular barriers to spread to neighboring cells (paracytophagy), causing diarrhea. *Shigella* sp. uses their type III secretion cells to inject several virulence factors in the host cells, causing membrane ruffling and macropinocytic uptake. The overexpression of dominant negative Arf6-N122I or Arf6 shRNA treatment of Hela cells or mouse embryonic fibroblasts significantly reduces *Shigella flexneri* internalization [91]. Arf6 and ARNO both accumulate at sites of *Shigella flexneri* entry, and Arf6 is activated in an early stage of infection by these virulent pathogens. The Shigella virulence factor **IpgD** is an inositol-4-phosphatase that converts PIP_2_ to phosphatidylinositol-5-phosphate (PI_5_P), which recruits PI3K and leads to PIP_3_ formation. IpgD and PI3K activity but, apparently, not Arf6 are required for ARNO recruitment while Arf6 recruitment and infection are inhibited by SecinH3, which blocks ARNO (and other related cytohesins). It was proposed that IpgD induces ARNO accumulation via PIP_3_, which in turn attracts and activates Arf6, leading to additional ARNO recruitment in a positive feedback loop [91]. While Arf6-GTP recruits ARNO in the model proposed for *Salmonella* Typhimurium (Figure 5B, [29,88]), ARNO recruits Arf6 in the model for *Shigella flexneri* (Figure 5C, [91]). The activation or recruitment of Arf1 by ARNO was not investigated in this setting, while Rac1 is activated by the Dock180/ELMO complex via the *Shigella* virulence protein IpgB1. Presumably, a combined Arf1/Rac1 activation leads to actin polymerization via the WRC.

***Yersina pseudotuberculosis*** and ***Yersinia enterocolitica*** are Gram-negative rods causing enterocolitis, yet they can migrate to other body parts and lead to diverse chronic inflammatory complications, such as arthritis. Infection by these *Yersinia* species takes place in the gastrointestinal tract, where they are able to translocate through M (microfold) cells and to enter Peyer’s patches for their replication, from where they can further disseminate to the lymphatic system. Entry into M cells requires the binding of the pathogen surface protein invasin to host cell β1 integrins [92]. In this mechanism of action, signaling via FAK, c-Src, PI3 kinase, Akt, and PLCγ1 all contribute to phagocytosis of the pathogen. In COS1 cells, Arf6 was localized at phagosomes containing *Yersina pseudotuberculosis* [93]. A high-affinity binding of *Yersinia* containing intact invasin was required to recruit Rac1, PIP5Kα, Arf6, and PIP_2_ onto nascent phagosomes. At these sites, Arf6 stimulates the activation of PIP5K and PIP_2_ production. The overexpression of Arf6 or Arf6-Q67L enhanced *Yersinia* uptake, while a nucleotide-free Arf6-N122I mutant or lowering PIP_2_ via a PIP_2_-specific phosphatase both reduced uptake [93]. An siRNA screen demonstrated that the Arf6 GAP **ADAP1** is needed for the accumulation of F-actin during the invasin-mediated uptake of *Yersinia enterocolitica*. ADAP1, the Arf6 GEF **Grp1**, and the Arf6 effector phospholipase D2 all localize at the site of entry [94].

The obligate intracellular bacterium ***Chlamydia caviae*** (another type III secretion system-containing pathogen) regulates its cellular uptake by inducing a transient Arf6 activation (5 minutes after bacterial contact). Arf6-induced activation of PIP5K and production of PIP_2_ at sites of bacterial entry are critical for pathogen uptake, while Arf6 activation of PLD is not required. The sites of uptake were enriched for F-actin and the Arp2/3 complex, creating membrane protrusions engulfing the bacterium [95].

***Listeria monocytogenes*** is a Gram-positive bacterial pathogen, causing gastroenteritis, meningitis, and abortion. Via its surface protein **InlB**, it interacts with and activates the Met receptor (also known as hepatocyte growth factor receptor). Subsequent cytoskeletal changes result in the uptake of *Listeria.* For *Listeria* entry and the associated cytoskeletal changes, Arf6 has to be converted to its GDP-bound form by the Arf6 GAP **ARAP2** [96].

While several invading pathogens described above activate Arf6 via host GEFs, the *Rickettsia and Legionella species* introduce the virulence factor **RalF**, which directly acts as a GEF for Arf6 and Arf1. ***Rickettsia typhi*** RalF is required for bacterial entry and induces a very early activation of Arf6, leading to PIP_2_ generation at the entry site, which is critical for invasion [97]. During a ***Legionella pneumophila*** infection, RalF is rather involved in recruiting Arf1 to the *Legionella*-Containing Vacuole after internalization, while *Legionella* RalF at the endoplasmatic reticulum may assist the trapping of host secretory vesicles [98].

Arf6 is also targeted in other aspects of bacterial virulence, such as **immune evasion**. Interference in host vesicular transport via Arf6 seems to be a common strategy in multiple pathogenic bacteria. **EspG** of EHEC and EPEC and **VirA** of *Shigella* sp. are structurally similar effectors of the type III secretion system. EspG is further shown to simultaneously bind Arf6 and p21-activated kinases (PAKs), blocking Arf-GAP activity in a GTP bound state and activating PAK [99]. VirA/EspG can also simultaneously bind Arf6 and Rab1 and deactivate Rab1 via their RabGAP domain [99,100]. EspG also binds Rab35 [101]. VirA/EspG-mediated tethering and the manipulation of Arf6 and other signaling/transport components leads to a drastic rewiring of endocytic traffic. EspG induces Golgi fragmentation, blocks the ER to Golgi traffic, and inhibits recycling via the endocytic recycling compartment, trapping receptors in an abnormal Rab11/Arf6 compartment [101]. EspG, thus, blocks the secretion of IL-8, preventing the recruitment, activation, and binding of immune cells. For *Shigella*, VirA contributes to its intracellular survival by counteracting an autophagy-mediated clearance of the intracellular pathogens (xenophagy) [100]. In this context, it is interesting to note that Arf6 is normally connected to the promotion of autophagy and may play a role in xenophagy (see later). 

The role of Arf6 in pathogen invasion is not limited to prokaryotic pathogens, as Arf6 is also abused by at least three **eukaryotic parasites**. Arf6 is required for the invasion of ***Toxoplasma gondii****,* which can cause toxoplasmosis. This obligate intracellular pathogen actively penetrates the host cell. Despite the active mechanism of penetration, an Arf6-T27N overexpression or Arf6 siRNA both reduce the degree of *Toxoplasma gondii* infection in Vero cells. In the host cell, *Toxoplasma* resides in a membrane-delineated compartment, the parasitophorous vacuole, which excludes host membrane proteins. However, Arf6 is recruited to the parasitophorous vacuole during invasion. Upon invasion, PI3K is activated, and it was suggested that this may activate Arf6, which in turn activates PIP5K. PIP5K activation induces PIP_2_ production, allowing the transportation of host cell membranes for vacuole formation to the site of parasite penetration [102]. The accumulation of Arf6 was also demonstrated on ***Trypanosama cruzi*** parasitophorous vacuoles in mouse embryonic fibroblasts, and ARF6 knockdown significantly lowered the number of internalized parasites [103]. The overexpression of Arf6-T27N strongly reduced an internalization of ***Leishmania donovani*** in RAW 264.7 macrophages [104]. 

## 4. Viruses Take Advantage of Arf6

Arf6 plays various roles in virus infections, as it can be both required and abused for virus entry, replication, and secretion but also the support restriction of virus infection. 

Arf6 is involved in **HIV-1** virulence during multiple stages, including viral entry, viral assembly and release, and immune evasion [105,106]. However, the importance of Arf6 in each of these steps is either debated or may be cell-type dependent. The silencing of Arf6 or the overexpression of Arf6-Q67L or T44N mutants inhibited the HIV-1 envelope induced-membrane fusion, entry, and infection of T lymphocytes [105]. The Arf6 mutants did not inhibit CD4-dependent virus attachment but blocked CD-4 dependent HIV-1 **internalization**. Arf6 knockdown also attenuated the cell-to-cell HIV-1 transmission of primary human CD4^+^ T lymphocyte [105]. This mechanism may not apply to every cell infected by HIV-1. Throphoblast cells (placental cells important for in utero HIV-1 transmission) display an unusual clathrin/caveolea-independent uptake of HIV-1 assisted by Rab5 and Rab7 without the apparent contribution of Arf6 [107]. HIV-1 was reported to use Arf6 for **virus assembly**. An overexpression of the constitutive active Arf6-Q67L mutant induces a formation of PIP_2_-enriched endosomal structures. HIV-1 **Gag**, a protein crucial for virus particle assembly, was redirected to these PIP_2_-positive vesicles and HIV-1 virions budded into these vesicles. Arf6 (in its function as PIP_2_-inducer) might, thus, regulate the plasma membrane localization of Gag, the place where HIV-1 virus assembly takes place in most cells [106]. Accordingly, the depletion of PIP_2_ via a phosphatase or Arf6Q-67L overexpression reduced virus production [106]. However, in another study, Arf6 depletion had no significant effect on HIV-1 particle release [108].

Via its role in CIE and recycling, Arf6 plays an important role in antigen presentation: Arf6 regulates the continuous endocytosis and recycling of both **MHC-I** and MHC-II molecules that are loaded with antigenic peptide [109,110,111]. The HIV-1 accessory protein **Nef** is used for HIV-1 immune evasion by downregulating immune receptors, including MHC-I and MHC-II at the cell surface of infected cells. For the downregulation of MHC-I, Nef was reported to target Arf6 [112]. Nef contains three motifs. Motif 1 interacts with the Golgi-associated protein PACS-1, and motif 2 is proposed to interact with PI3K. Both motifs 1 and 2 are required for an enhanced endocytosis of MHC-I, and the co-expression of Nef with PACS-1 leads to Arf6 activation to its GTP bound form [112]. The third motif was essential for the transportation of internalized MHC-I to the Golgi apparatus and for a reducing recycling of MHC-I to the plasma membrane. A model was proposed where a Nef-PACS1 complex activates PI3K, leading to an increased ARNO association, activating Arf6 to internalize MHC-I, followed by rerouting to the Golgi apparatus [112]. Two other studies could not confirm an involvement of Arf6 in the Nef downregulation of MHC-I in T-cells [113,114]. The role of Arf6 in Nef-mediated MHC-I downregulation was confirmed in a very extensive study, demonstrating that the Arf6 requirement was cell-type dependent [115]. While Nef can affect MHC-I recycling via Arf6, it can also reduce MHC-I cell surface expression by stabilizing the interaction of MHC-I with AP-1, leading to a lysosomal degradation of newly synthesized MHC-I. Both pathways are used to different extents in different cells [115].

**Coxsackieviruses** cause infections of the central nervous system, myocarditis, and respiratory diseases. 

Arf6 seems to play distinct roles during the infection of different Coxsackieviruses. Coxsackievirus A9 binds to integrin αVβ3 but needs β2-microglobulin, a subunit of the MHC-I complex for its internalization. Arf6 is important for the internalization and recycling of MHC-I and integrins and affects the internalization of Coxsackievirus A9: Arf6 siRNA slightly inhibits infection, but the overexpression of Arf6-T27N caused an 85% reduction in infection [116,117]. In contrast, Arf6 seems to be involved in the **restriction** of a Coxsackievirus B3 infection. The virus internalizes by binding to the Coxsackie-adenovirus receptor, activating ERK, which in turn triggers Arf6 via an unknown mechanism and leads to a relocalisation of the virus to a nonproductive compartment [118].

The Arf6 GAP **ADAP2** is also involved in **virus restriction**. The ADAP2 expression is upregulated by type I interferons, and an ADAP2 overexpression restricts infection by dengue virus (DENV) and vesicular stomatitis virus (VSV) via a mechanism that requires intact Arf6 GAP activity [119]. ADAP2 expression induces macropinocytosis and restricts DENV infection during virion entry and/or intracellular trafficking. Incoming DENV and VSV particles already associate with ADAP2 at their entry, which may affect their proper internalization and transport via Arf6.

**Cytomegalovirus** (CMV) rearranges the Golgi compartment and probably most endosomal transport systems to generate a juxtanuclear virus assembly compartment. An infection alters the fate of cargo internalized via clathrin-independent and clathrin-mediated endocytosis. In a study with human CMV, cargo accumulates in endosomal structures associated with the sorting endosome marker EEA1, which display an increased concentration of Arf6 [120]. Arf6 in infected cells show a decreased association with GTP, and it was suggested that Arf6 mediated recycling is hampered in infected cells. In a study using murine CMV, endocytic cargos destined for recycling or late endosomes both accumulate in structures enriched in Arf6, Rab11, Rab22, and Rab5 [121]. Both studies show that endocytic recycling is inhibited, with a possible role for Arf6. The CMV **glycoprotein M** was shown to interact with FIP4 and to block the FIP4-Arf6 interaction but not the FIP4-Rab11 interaction, which may play a role in the altered endocytic recycling [122].

**Vaccinia viruses** enter cells by a mechanism that resembles macropinocytosis. The overexpression of Arf6-Q67L inhibits this macropinocytosis process and reduces infectivity by 78% [123]. However, another study could not confirm a role for Arf6 in a Vaccinia infection [124]. The vaccinia protein K1L is required for virus replication in certain cell lines, and the deletion of specific ankyrin repeats in K1L affects host-range specificity. A critical ankyrin repeat interacts with the Arf6 GAP ACAP2, but it is unknown whether ACAP2 contributes to this host-range specificity, as ankyrin repeat mutations that affect viral replication in HeLa cells did not affect binding of ACAP2, pointing to a possible other function of the ankyrin repeats in virus replication [125,126].

The orally transmitted **Epstein-Bar virus** was shown to cross oral epithelial cells via bidirectional transcytosis. Apical to basolateral transcytosis was blocked by the inhibition of proteins that typically mediate macropinocytosis, including the siRNA silencing of Arf6 [127].

## 5. Arf6 Activation Precedes Reactive Oxygen Species (ROS) Production

Via specialized **NADPH oxidase** complexes, phagocytic cells and especially neutrophils can release large quantities of reactive oxygen species to kill pathogens [128]. When ingesting pathogens via phagocytosis, the activation of the Fc receptors, complement receptors, or Toll-like receptors triggers ROS release in the phagosome, killing the ingested pathogen. Neutrophils also release extracellular ROS when triggered by IL-8 or by N-formyl oligopeptides released by bacteria, such as formyl-methionine-leucine-phenylalanine. Arf6 appears to be crucial in activating the NADPH oxidase complex via its activation of phospholipase D. Arf6 was required for the strongly increased NADPH oxidase activity in neutrophil-like PLB-985 cells following stimulation with formyl-methionine-leucine-phenylalanine (fMLP). PLD was activated by Arf6 and the induced activation of NADPH oxidase [129]. These findings were confirmed by a novel benzyl indazole compound (CHS-111) inhibiting fMLP-induced PLD activity by attenuating the interaction between Arf6 and PLD [27]. Another compound (Fal-002-2) inhibiting superoxide anion radical formation in fLMP-stimulated rat neutrophils functions by blocking the activation of Arf6 and the interaction between Arf6 and PLD1 [130]. In this process, the activation of Arf6 is mediated by the GEF cytohesin-1 [131]. Similarly, the Fcγ receptor activation of NADPH oxidase requires a PLD1 activation, and Fcγ receptor signaling activates PLD via Arf6 [132]. 

## 6. The Crucial Role of Arf6 in TLR Signaling

It is becoming more and more evident that Arf6 is critical for the signaling pathways of the Toll-like receptors. The innate immune receptor TLR4 recognizes the lipopolysaccharides (LPS) of Gram-negative bacteria. TLR4 signaling goes via a Myeloid differentiation factor 88 (**MyD88**)-dependent and a TIR-domain-containing adaptor-inducing interferon-β (**Trif**)-dependent pathway (Figure 6). The former pathway starts with the recruitment of a MyD88-adaptor like (**Mal**) to the plasma membrane, where it links MyD88 to the activated TLR4, resulting in the activation of the transcription factors nuclear factor κB (NF-κB) and AP-1, promoting the transcription of proinflammatory cytokines [133,134] The Trif-dependent pathway requires an internalization of TLR4, upon which the adaptor Trif-related adaptor molecule (**Tram**) binds to TLR4, enabling the recruitment of Trif [135]. This results in an activation of interferon-regulator factor 3 (IRF3) and the production of type I interferons. An Arf6 inhibitory peptide was shown to suppress the LPS-induced production of the chemokine KC in mouse embryonic fibroblasts [20]. An overexpression of Arf6-T27N redistributed Mal to Arf6-containing vesicles, and Arf6 was suggested to function in TLR4 signaling by activating PIP5K, which induces PIP_2_ production. Via its PIP_2_-binding domain, Mal is recruited to the activated TLR4 where it initiates the MyD88-dependent signaling pathway [20] (Figure 6). However, another study failed to observe an LPS-induced activation of Arf6 and could neither demonstrate a role of Arf6 in LPS-induced bone marrow-derived dendritic cell maturation nor demonstrate a role in LPS-induced cytokine production [136]. Using an Arf6 blocking peptide, the overexpression of Arf6 mutants, and Arf6 shRNA, we confirmed the necessity of Arf6 in TLR4 signaling in different cell types and demonstrated that Arf6 is essential for both the MyD88-dependent and MyD88-independent TLR4 pathways [19]. Interestingly, the Arf6 inhibitory peptide still affects a LPS-induced TNF-α secretion in Mal-KO macrophages, suggesting that it affects MyD88-dependent signaling via mechanisms independent of PIP_2_-dependent Mal recruitment. By controlling the internalization of LPS and the subsequent transport of Tram to the internalized TLR4, Arf6 also controls IRF3 phosphorylation and IRF3-dependent gene production [19] (Figure 6). LPS stimulation leads to a fast and transient increase in the Arf6-GTP concentration, suggesting that LPS is able to activate an unknown Arf6 GEF. The fast response indicates that the TLR4 MyD88- and TRIF-dependent signaling pathways may be dependent on an earlier LPS-induced Arf6 activation step.

Like TLR4, the plasma membrane-expressed TLR2 and its heterodimer complexes (TLR1/2 and TLR2/6) initiate the MyD88-dependent pathway by using Mal as a bridging adapter [137,138]. Unexpectedly, using an Arf6 blocking peptide, Arf6 shRNA, and ARf6 mutant overexpression, we found that the TLR2-dependent NF-κB activation is independent of Arf6 in THP-1 cells or HEK293-TLR2 cells when stimulated with the TLR2 ligand lipoteichoic acid, the TLR2/6 ligand PAM2CSK4, or the TLR1/2 ligand PAM3CSK4 [19]. 

A MyD88-dependent NF-κB activation by the endosomal TLR9 is strongly dependent on Arf6, as Arf6 regulates the internalization of its ligand unmethylated CpG DNA [139,140]. A basal level of activated Arf6 is required for the uptake of oligonucleotides, and internalized CpG stimulates TLR9 [140]. This leads to a subsequent enhanced activation of Arf6, which increases the specific uptake of CpG oligonucleotides. This Arf6 activation requires PI3K activity, presumably via a TLR9-PI3K-Arf6 pathway [139]. Arf6 is also required for the trafficking of TLR9 to endolysosomes and subsequent TLR9 proteolytic processing [139].

The role of Arf6 in TLR signaling was confirmed in a study that demonstrated an essential role for Arf6 and the **V-type ATPase** in TLR responses [141]. In this study, ARF6-KO RAW264.7 mouse macrophage cells show a reduced LPS-induced activation of NF-κB and IRF3 and reduced LPS-induced IL-6 and interferon-β expression. The ARF6-KO RAW264.7 cells showed an increased surface expression of TLR4, which indicates a role of Arf6 in TLR4 internalization. Moreover, the cells did not show LPS-induced TLR4 endocytosis which occurs in normal macrophages. The induction of IL-6 by the TLR2/TLR6 ligand MALP-2 is also reduced in these cells, indicating that Arf6 does play a role in TLR2/TLR6 signaling.

In this study, the effects of Arf6 were linked to the interaction of Arf6 with the multimeric V-type ATP-ase [141]. Arf6 interacts with the c subunit of the V-type ATPase, while ARNO interacts with the a subunit [141,142,143]. The V-type ATPase complex is a proton pump that is required for the acidification of endosomes, which is important for the degradation, transport, and processing of proteins; for breaking ligand-receptor contacts in endosomes; and for the inactivation of internalized pathogens. The recruitment of Arf6 and ARNO to endosomes is driven by the V-type ATPase and is dependent on intra-endosomal acidic pH. An inhibition of the V-type ATPase affects vesicular transport and, thus, has an indirect delayed inhibitory effect on the endocytosis of cargo [144]. It does not seem to inhibit the transport of internalized transferrin to the Rab11+ compartment but blocks transport from early to late endosomes. Cells deficient in the V-ATPase subunit ATP6V0D2 display a reduced cytokine response after stimulation of the endosomal TLR3, TLR7, and TLR9. In contrast, LPS-induced inflammatory cytokine production and NF-κB activation was increased. The cells had defects in the internalization of cell surface TLR4 and in LPS tolerance, as they did not display the normal decrease in LPS-induced cytokine production after repeated LPS simulation. In line with defective TLR4 internalization, the cells were defective in LPS-induced IRF3 activation.

These finding suggest that TLR4 internalization, required for TRIF-dependent signaling and possibly for LPS tolerance, requires both Arf6 and V-ATPase and that Arf6 and V-ATPase may be mechanistically linked in this process, although the exact mechanism remains unclear [141]. The assembly of the V-ATPase increases during maturation of bone marrow-derived dendritic cells induced by activation of Toll-like receptors [145]. This assembly is essential for antigen processing, which requires an acidic lysosomal pH. This suggests an interesting mutual interdependence between TLRs and the V-ATPase, which is possibly linked via Arf6 and PI3K.

miR-145 downregulates Arf6 expression, and miR-145 overexpression in THP-1 reduces the LPS-induced NF-κB activation and release of IL-1beta, TNF-α, and IL-6 [146]. The protein AIP1 was proposed as an Arf6 GAP that affects TLR4 signaling, but the identification of AIP1 as an Arf6 GAP was later questioned and retracted. ARNO was shown to act as an Arf6 GEF in TLR4- and IL-1-mediated processes. Like TLRs, the IL-1 receptor contains an intracellular Toll/Interleukin-1 receptor (TIR) domain and uses MyD88 for its signal transduction (Figure 6). Both IL-1 and LPS are reported to induce a MyD88-ARNO-Arf6 pathway in vascular endothelial cells to reduce vascular endothelium cell–cell interaction and to induce vascular leaks [147,148]. A direct MyD88-ARNO interaction was shown via co-immunoprecipitation. In these studies, Arf6 siRNA and an Arf6 inhibitory peptides do not affect the LPS-induced NF-κB activation of TNF-α release from vascular endothelial cells, which seems to be at odds with other studies that demonstrate the effects of Arf6 inhibition/modulation in multiple other cell types [19,20,141]. The SecinH3 inhibitor of ARNO inhibits the IL-1-induced Arf6 activation to Arf6-GTP and increases vascular stability [147]. We could not find any effect of SecinH3 on LPS-induced cytokine secretion or NF-κB activation, while four other types of Arf6 modulation/inhibition clearly affected these readouts. This indicates that Arf6 but not the Myd88-ARNO-Arf6 pathway is important for classical MyD88 and TRIF-dependent TLR4 signaling to IRF3 and NF-κB, and the Arf6 GEFs and GAPs that affect these pathways remain to be identified.

Arf6 may be related to several other effects induced by TLR activation. In dendritic cells, TLR4 signaling by bacterial components trapped in phagosomes triggers the transport of MHC-I from a Rab11 positive juxtanuclear compartment to the phagosome to enable the loading of phagosomal peptides on the MHC-I for cross presentation [149]. Conversely, Rab11 was shown to regulate the transport of TLR4 from the ERC towards phagosomes, and this is essential for TRIF-dependent signaling [135]. Given the role of Arf6 in MHC-I recycling, TLR4 signaling, and the TRIF dependent pathway and since Arf6 and Rab11 often cooperate, it is tempting to speculate that Arf6 may also play a role in these transport processes. The detection of pathogen-associated molecular patterns (PAMPs) in phagosomes by Toll-like receptors (TLRs) leads to a recruitment of autophagy markers, which enhances the lysosomal degradation [150]. As Arf6 is implicated in autophagy [60,61], the TLR activation of Arf6 may play a role in this process. 

## 7. Conclusions

Arf6 is clearly important for many aspects of host–pathogen interactions and innate immunity. It has an important role in phagocytosis, TLR signaling, and ROS production but also in later aspects of immune defense such as antigen presentation. Arf6 is also targeted or hijacked by pathogens to enter host cells or to evade the immune response. While the involvement of Arf6 is evident, many open questions remain. In most examples highlighted in this review, it is unclear how Arf6 is activated or inactivated, as the exact triggers that recruit or activate a GEF or GAP or even the identity of these GAPs or GEFs are often unclear or unknown. Arf6 is involved in TLR signaling, phagocytosis, pathogen invasion, ROS production, and MHC-I recycling, which all converge at the phagosome, but it remains unclear how/whether the activation(s) and effects of Arf6 in these mechanisms overlap or whether Arf6 links these processes together in an integrated mechanism. Arf6 interacts with adapters FIP3 and 4 and JIP3 and 4 that couple to motor proteins, and the role of this interaction has been studied in depth in its role in cytokinesis. The potential role of this interaction and its regulation in host–pathogen interaction and immunity are also completely unknown. 

Blocking Arf6 in specific processes has been proposed as a potential target for therapeutic intervention to block cancer cell invasion and other aspects of oncogenesis and to improve endothelial barrier function in endotoxaemia and heart failure. It could also be targeted to inhibit the invasion and survival of pathogens or to suppress excessive immune responses. However, Arf6 itself and most of its GEFs are involved in a large number of important cellular processes. Further research into the details of the processes that lead to Arf6 activation or the exact effects provoked by Arf6 in specific contexts may lead to new therapeutic opportunities.

## Figures and Tables

**Figure 1 ijms-20-02209-f001:**
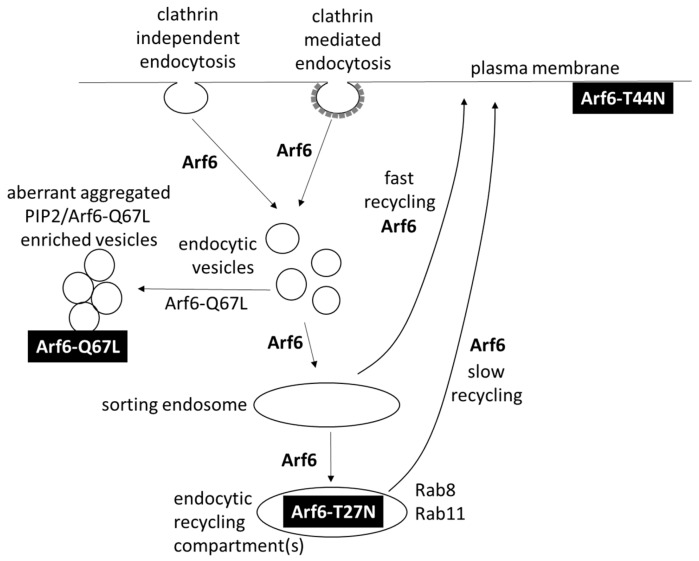
The role of Arf6 in intracellular trafficking upon clathrin-mediated endocytosis and clathrin-independent endocytosis. The location of Arf6 mutants in the cell is indicated as the white text on a black background.

**Figure 2 ijms-20-02209-f002:**
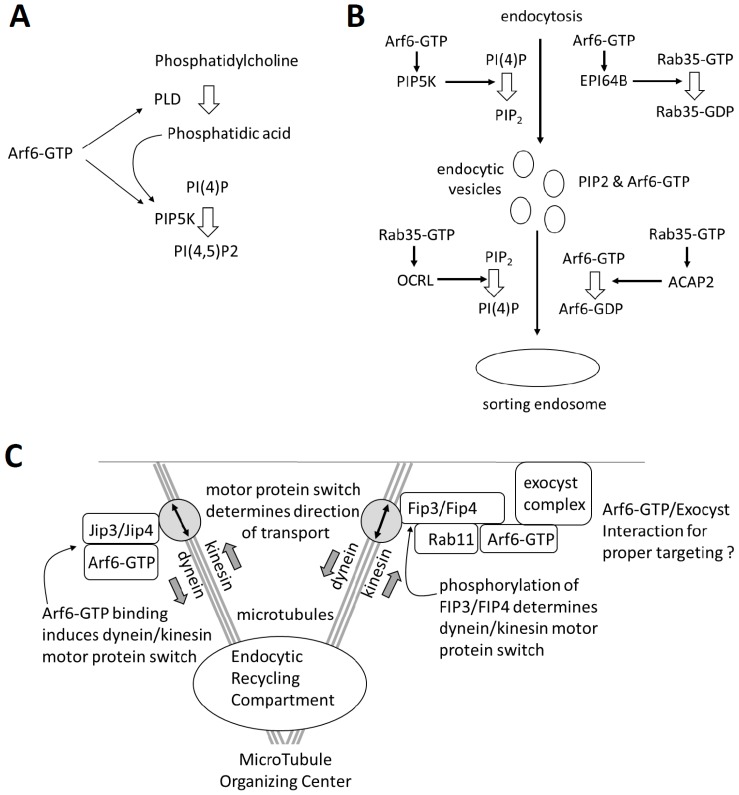
A schematic overview of some of the major effectors of Arf6: (**A**) Arf6 activates phosphatidylinositol 4-phosphate 5-kinase (PIP5K) and phospholipase D (PLD). (**B**) The antagonism between Arf6 and Rab35 is shown here in the context of early steps of vesicular transport after endocytosis. (**C**) The Arf6 interaction with adapter proteins Jip3/4, Fip3/4, and the exocyst.

**Figure 3 ijms-20-02209-f003:**
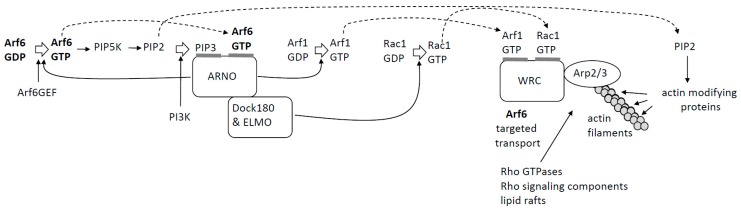
A schematic overview of Arf6’s role in actin polymerization. Dashed arrows indicate that the product is used in another step.

**Figure 4 ijms-20-02209-f004:**
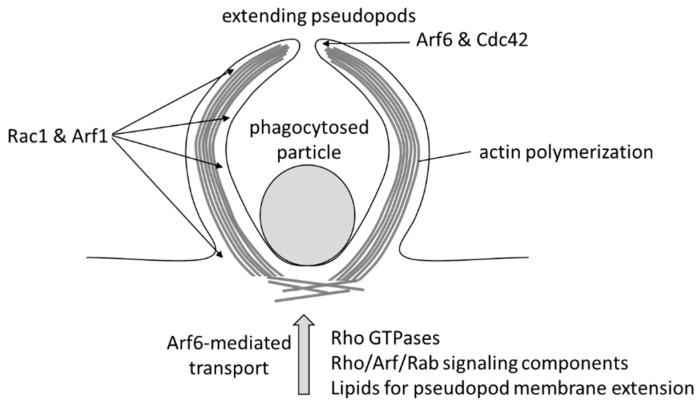
Arf6 in the phagocytic cup: Arf6 and Cdc42 are activated in an early stage of phagocytosis at the tips of the extending pseudopods. Rac1 and Arf1 show a broader distribution along the pseudopods. Arf6 is also responsible for the transport of vesicles to the phagosome.

**Figure 5 ijms-20-02209-f005:**
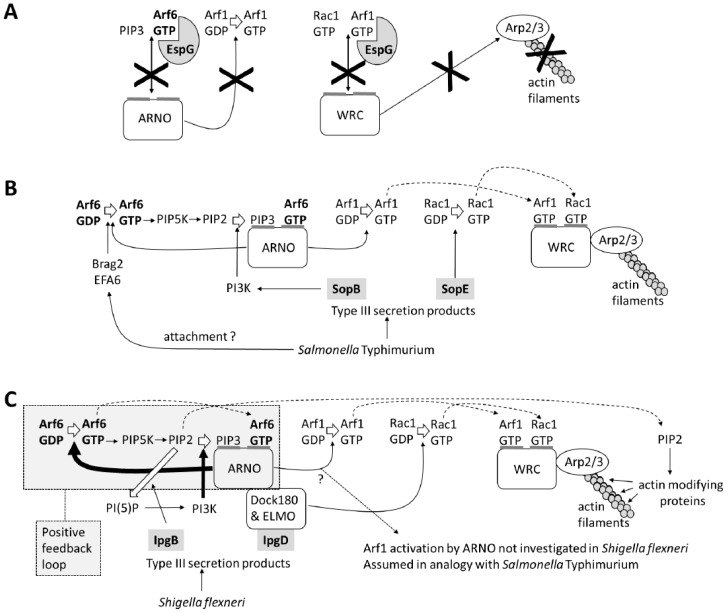
A subversion of Arf6 by bacteria: (**A**) EspG of enteropathogenic and enterohemorrhagic *Escherichia coli* (EHEC and EPEC) blocks the interactions of Arf6 and Arf1 to inhibit actin polymerization. *Salmonella* Typhimurium (**B**) and *Shigella flexneri* (**C**) use Arf6 and its guanine nucleotide exchange factors (GEF) ARNO to promote actin polymerization. Dark crosses indicate that the process is blocked by EspG. Dashed arrows indicate that the product is used in another step. Thick arrows indicate that the process is amplified by IpgB or the positive feedback loop.

**Figure 6 ijms-20-02209-f006:**
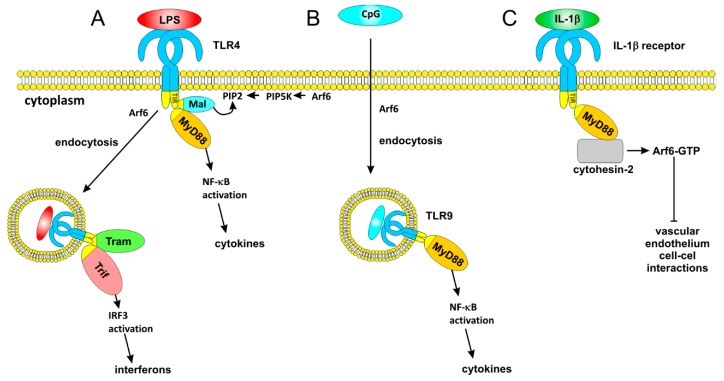
The role of Arf6 in TLR signal transduction. (**A**) TLR4 requires Arf6 for its MyD88-dependent and Trif-dependent signaling. Arf6 promotes PIP2 formation via PIP 5-kinase. The newly formed PIP2 helps the binding of Mal, which contains a PIP2-binding domain. Mal helps to recruit MyD88 to the activated TLR4. The Trif-dependent signaling requires endocytosis of the activated receptor complex. Arf6 is required for the internalization of LPS and the transportation of Tram and may be required for an internalization of the TLR4 complex. (**B**) TLR9 is present in endosomes. Arf6 is required for the internalization of its ligand, CpG DNA. (**C**) Although the IL-1β receptor also signals via MyD88, Arf6 is not required for its MyD88-dependent signaling. However, Arf6 is required for an IL-1β-induced vascular instability. This function is mediated by a direct interaction of the Arf6 GEF cytohesin-1.

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
