# Peer review of "The Small GTPase Arf6: An Overview of Its Mechanisms of Action and of Its Role in Host–Pathogen Interactions and Innate Immunity"

_ijms, 2019, doi:10.3390/ijms20092209_

Round 1
Reviewer 1 Report
This is a well-written review on Arf6 role in host-pathogen interactions and innate immunity. The Arf6 involvement of various cellular pathways is nicely illustrated in the figures. There are several typos in the manuscript that need to be corrected before accepting it.
Minor corrections
Pg1, Line 13: Change 'PI5 kinase' to 'PIP 5-kinase'
Pg1, Line 29: Change 'in' to 'into'
Pg1, Line 36: Change from 'the membrane' to 'the plasma membrane'
Pg1, Line 37: Change from 'the membrane' to 'the plasma membrane'. Since Arf6-GDP is also known to localize to endosomes, it may not be correct to say 'dissociation from the membrane requires GTP hydrolysis on Arf6 (see Figure 1).
Pg2, Line 39: Either define them or mention them in brackets next to clathrin-independent endocytosis (CIE) and clathrin-mediated endocytosis (CME) in the figure.
Pg2, Lines 62-63: The sentence needs re-writing for the clarity
Pg2, Lines 66-67: Incorrect, it displaces Arf6 from the membrane
Pg3, Line 71: Change 'PI5K' to 'PIP5K'
Pg3, Line 85: Change 'Activate' to 'Active' or 'Activated'
Pg4, Line 99: Change 'end' to 'and'
Pg10, Line 365: Change 'HIV-1: throphoblast cells' to 'HIV-1. Trophoblast cells'
Pg11, Line 413: Change 'EEA6' to 'EEA1'
Pg15, Line 569: Change 'is it' to 'it'
Author Response
We thank the reviewers for carefully reading the MS and for the positive comments. We applied all suggested corrections on the MS with one exception: "Pg2, Lines 66-67: Incorrect, it displaces Arf6 from the membrane". We did change the sentence a little bit to :"The peptide acts as an antagonist of Arf6, possibly via competitive binding to Arf6 interaction partners, although the exact targets of this peptide have not been defined (to our knowledge)."
Minor corrections
Pg1, Line 13: Change 'PI5 kinase' to 'PIP 5-kinase' OK (we also changed PI5 kinase in other sites in the text and figures)
Pg1, Line 29: Change 'in' to 'into' OK
Pg1, Line 36: Change from 'the membrane' to 'the plasma membrane' OK
Pg1,
Line 37: Change from 'the membrane' to 'the plasma membrane'. Since
Arf6-GDP is also known to localize to endosomes, it may not be correct
to say 'dissociation from the membrane requires GTP hydrolysis on Arf6
(see Figure 1). OK
Pg2, Line 39: Either define them or mention them
in brackets next to clathrin-independent endocytosis (CIE) and
clathrin-mediated endocytosis (CME) in the figure. OK
Pg2, Lines 62-63: The sentence needs re-writing for the clarity OK
Pg2, Lines 66-67: Incorrect, it displaces Arf6 from the membrane
We changed the sentence to: "The peptide acts as an antagonist of Arf6, possibly via competitive binding to Arf6 interaction partners, although the exact targets of this peptide have not been defined (to our knowledge)."
Pg3, Line 71: Change 'PI5K' to 'PIP5K' OK
Pg3, Line 85: Change 'Activate' to 'Active' or 'Activated' OK
Pg4, Line 99: Change 'end' to 'and' OK
Pg10, Line 365: Change 'HIV-1: throphoblast cells' to 'HIV-1. Trophoblast cells' OK
Pg11, Line 413: Change 'EEA6' to 'EEA1' OK
Pg15, Line 569: Change 'is it' to 'it' OK
Reviewer 2 Report
This manuscript summarizes the current knowledge of the role ARF6 plays in host-pathogen interactions and innate immunity.
It is well written, easy to read and contains very nice illustrations facilitating comprehension. The authors have cited properly the published works related to their topic of interest. I only found a few spelling/editing mistakes that should be corrected.
Overall, very nice review.

Author Response
We thank the reviewers for carefully reading the MS and for the positive comments. We applied all suggested corrections on the MS.